# The relationship between health literacy and the adoption of COVID-19 preventive behaviors: A cross-sectional study in Iran

Rahman Panahi[1], Zahra Ghorbanpour[2], Bagher Moradi[2], Fereshteh Eidy[3], Mohiadin Amjadian[4]*

1 Department of Public Health, Qaen School of Medical Sciences, Birjand University of Medical Sciences, Birjand, Iran, 2 Esfarayen Faculty of Medical Sciences, Department of Public Health, Esfarayen, Iran, 3 Faculty of Medical Sciences, Department of Biostatistics, Tarbiat Modares University, Tehran, Iran, 4 English Language Department, School of Medicine, Kurdistan Medical Sciences University, Sanandaj, Iran

* mohiadin72@yahoo.com

**Data Availability Statement:** All relevant data are within the paper and its Supporting Information files.

## Abstract

### Background and aim

Health literacy (HL) is one of the effective factors in controlling the COVID-19 epidemic. Considering the high prevalence of COVID-19 disease, the present study aimed to determine the relationship between HL and the adoption of COVID-19 preventive behaviors.

### Materials and methods

This was a descriptive-cross sectional study conducted on 214 students selected by simple random sampling method in Esfarayen Faculty of Medical Sciences, Iran, in 2022. The data included demographic and background characteristics, health literacy for Iranian Adults (HELIA), and a valid and reliable questionnaire to measure COVID-19 preventive behaviors. The data were analyzed using SPSS 23, descriptive statistics, and ordinal logistic regression tests.

### Results

The mean and standard deviation of the scores of adoption of COVID-19 preventive behaviors and HL among students were 18.18) 4.02(out of 25 and 72.14) and 12.75 (out of 100, respectively. The results of the logistic regression test showed that the HL (P = 0.003), gender (P<0.001), mother's education (P = 0.039), educational level (P = 0.031), smoking (P = 0.032), and physical activity (P = 0.007) were effective factors in adopting preventive behaviors.

### Conclusion

Adopting preventive behaviors against COVID-19 was lower among students with lower levels of health literacy, male students, students with less physical activity, students with illiterate mothers, undergraduate students, and finally smokers. Therefore, it is necessary to pay more attention to these students in designing educational programs. It is suggested to carry

**Funding:** he author(s) received no specific funding for this work.

**Competing interests:** The authors have declared that no competing interests exist.

**Abbreviations:** HL, Health literacy; HELIA, Health Literacy for Iranian Adults; SD, Standard deviation; DE, Design effect; CVI, Content validity index; CVR, Content validity ratio.

out more extensive studies to clarify the effect of HL on the adoption of COVID-19 preventive behaviors.

## Introduction

Today, the coronavirus (SARS-COV-2) is considered a dangerous viral infection for public health [1]. Although most severe complications and mortality were seen in elderly people with background diseases such as liver and kidney diseases, and cancer [2], the risk exists for all people, and the importance of prevention is highlighted because of asymptomatic carriers who can transmit the disease to high-risk people and cause higher rates of mortality [3]. Measures such as education, improving awareness and attitude, and adopting preventive actions to protect against coronavirus disease are important strategies for prevention [4]. The World Health Organization considers washing hands regularly, respiratory hygiene, maintaining proper physical distance, and avoiding shaking hands and hugging as important behaviors to prevent this disease [5].

To identify preventive measures and control viral diseases, factors affecting preventive behaviors against respiratory viral diseases should be identified [6]. One of the effective factors in controlling the epidemic of Coronavirus Disease (COVID-19) is health literacy [7]. Also, health literacy is related to the adoption of preventive behaviors [8]. Health literacy is a cognitive skill and an influential issue in the healthcare system, for which various definitions have been provided so far. Health literacy is defined as a wide range of knowledge and skills for acquiring, processing, and applying health information [9].

Fortunately, low health literacy can be promoted by educating people [10]. Studies show that many diseases are associated with insufficient health literacy, so health literacy has a greater impact on people's health than other variables such as income, employment status, age, educational level, and race; because people with low health literacy do not understand the written or spoken information of the health team and will not act on it. As a result, they have a poor health status and spend more money to improve their health [9–11].

The university student population in Iran has grown significantly in recent years [12]. Due to the important role they play as managers and future planners in the country [13], students can contribute to having a healthier society by promoting their health literacy [12]. Therefore, it is very important to know the factors that are effective in adopting healthier behaviors among students to reduce their risky behaviors [12]. The results of Sajjadi et al.'s study showed that one-third of the students had insufficient or not very sufficient health literacy [12]. Moreover, the health literacy of students was reported to be in a moderate level in the study of Vozikis et al. [14]. Therefore, Students were chosen to be studied because of their suitable age [13], learnability, as well as being a suitable model for a healthy lifestyle [13,15].

Considering the outbreak of COVID-19 disease [4], the role of health literacy in controlling COVID-19 [7], adopting preventive behaviors [8], and the role of students in promoting a healthier society [13], this study aimed to determine the relationship between health literacy and the adoption of preventive behaviors in the COVID-19 epidemic in university students.

## Materials and methods

This was a cross-sectional descriptive-analytical study conducted between January and April 2022. The population included all male and female students at Esfarayen Medical Sciences University in Iran. The names of all students were listed and 214 participants were randomly

selected by simple random sampling. The required sample size was estimated to be 175 based on the results of the pilot study among 30 students (considering r = 0.15 for the correlation between health literacy and adoption of preventive behaviors against COVID-19 disease), and also using the sample size table for correlational research [16]. Then, by considering the design effect (DE = 1.2), the sample size was estimated to be 210. Finally, with the possibility of a 5% dropout of samples, 220 students were included in the study.

The inclusion criteria included studying at the Esfarayen Medical Sciences University, willingness to participate in the study, having informed consent to participate in the study, being in the age range of 18 to 65 years, and having Iranian citizenship. Unwillingness to continue the study and incomplete questionnaires were considered exclusion criteria.

The data were gathered by three questionnaires: 1- A demographic and background information questionnaire, 2- Health literacy for Iranian Adults (HELIA), and 3- A questionnaire to measure the adoption of preventive behaviors against COVID-19.

The first part included items about the participants' gender, years of education, marital status, parents' occupation, parents' education, smoking, hookah use, place of residence, level of education, field of study, amount of physical activities per week, method of receiving information, and type of COVID-19 infection.

In the second part, to measure health literacy, a health literacy questionnaire for the Iranian urban population between 18 to 65 years old (HELIA) was used [17]. This questionnaire consisted of 33 items in 5 areas, including access (6 items), reading skill (4 items), comprehension (7 items), evaluation (4 items), and decision-making and use of health information (12 items). The scoring scale was a five-option Likert scale that ranges from 5 (completely easy or always) to 1 (completely difficult or never). The raw score of every participant in each area was obtained from the algebraic sum of scores. To convert this score to a range of 0 to 100, the difference of the raw score obtained from the minimum possible raw score divided by the difference of the maximum possible score from the minimum score was used. Then, to calculate the total score, the scores of all the dimensions (based on the range from zero to 100) were added, and divided by the number of dimensions (number 5). Scores from 0 to 50 were considered inadequate health literacy, 50.1 to 66 as insufficient health literacy, 66.1 to 84 as adequate health literacy, and 84.1 to 100 as excellent health literacy [17]. This questionnaire has been validated by Mantazeri et al., and Panahi et al. in the general population and university students, respectively, and its validity and reliability were confirmed [18,19]. Also, Cronbach's alpha coefficient was acceptable for the dimensions of reading ($\alpha$ = 0.84), access ($\alpha$ = 0.85), understanding ($\alpha$ = 0.90), evaluation ($\alpha$ = 0.77), decision-making, and use of health information use ($\alpha$ = 0.86), and the entire questionnaire ($\alpha$ = 0.94).

In the third part, the questionnaire designed by Khazaei-Poul et al. [4] was used. This questionnaire contained 5 items and the range of its score was 5 to 25. The options (never-rarely-sometimes-often-always) were given a score of 1 to 5 respectively. All validity (Content validity ratio and content validity index) and reliability steps of this scale have been done and confirmed [4]. Also, in the present study, the questionnaire was pilot-studied among 30 university students, and its Cronbach's alpha coefficient was calculated to be 0.84. The rate of adoption of preventive behaviors was classified into 3 levels: poor (a score less than 50% of the total score), medium (a score of 50–75% of the total score), and good (a score above 75% of the total score) [20]. However, according to the researcher's opinion, the rate of adoption of preventive behaviors was classified into 2 levels: poor (scores less than 50% of the total score: less than 12.5) and good (scores more than 50% of the total score: more than 12.5) [21–24] and then it used in the logistic regression.

After receiving the code of ethics from the Deputy of Research and Technology of Esfarayen Medical Sciences University (IR.ESFARAYENUMS.REC.1401.009) explaining the objectives

of the study to the participants, and receiving their written informed consent, the questionnaires were given to the participants to fill out. The students were assured that their information would be used confidentially. In addition, the questionnaires were self-reported. Then the data were analyzed using descriptive statistics (Mean, standard deviation, frequency, and percentage) and ordinal logistic regression by SPSS version 23. The variables were entered simultaneously, and by comparing the independent variables classified as indicators, the last class of variables was selected as the reference class. The significance level was considered to be less than 0.05.

## Findings

A total of 214 students of the Esfarayen Universit of Medical Sciences were studied (response rate: 97.3%). Among them, 57.5% (123) of the samples were women, 89.3% (191) were single and 79% (169) lived in the dormitory, 54.8% (128) were studying in clinical fields. 74.8% (160) of the students stated that they were infected with COVID-19 (Table 1). Also, the average and standard deviation of the score of adoption of preventive behaviors related to COVID-19 and health literacy among students were 18.18±4.02 out of 25 and 72.14±12.75 out of 100, respectively (Table 2).

The results showed that there was a significant relationship between health literacy, gender, mother's education, educational level, smoking, and the amount of sports activity of students with the adoption of preventive behaviors against COVID-19.

Participants with inadequate, and not very adequate health literacy adopted less preventive behaviors by 90% and 66%, respectively, compared to the participants with excellent health literacy. (p-value <0.05). Male students adopted preventive behaviors 75% less than the female ones (p-value <0.001). However, students whose mothers had non-university education adopted these behaviors 2.25 times more than the students whose mothers had a university education (p-value <0.039). Associate degree students adopted these behaviors 3.02 times more than undergraduate ones (p-value <0.031). Also, the smoking students adopted preventive behaviors 70% less than the other students (p-value <0.032). Moreover, the students with high and moderate levels of sports activities adopted these behaviors 76% and 57% less than the students with low levels of sports activity respectively (p-value <0.05) (Table 3).

## Discussion

This study aimed to determine the relationship between health literacy and the adoption of preventive behaviors against COVID-19 among students. The results showed that health literacy level in the students was moderate. Considering the effect of the level of education on the level of health literacy of people [12], and the fact that almost two-thirds of the students in the present study were in the first and second years of study, these results were justifiable to some extent. These results were consistent with the results of Panahi et al. [24], and Panahi et al. [25] which were conducted among university students to determine the relationship between health literacy and the adoption of preventive behaviors. However, they were not consistent with the results of Sajjadi et al. [12] who reported a high level of health literacy in university students. The students' higher educational levels in their study might be the possible reason for the difference from the present study. It is worth noting that the aforementioned studies were conducted in university students by HELIA questionnaire [12,24,25]. In addition, the results of the present study were not consistent with the results of the study by Vozikis et al. in which the level of health literacy of students was reported to be medium to high [14]. Among the possible reasons for this discrepancy, we could point out the difference in the measurement tool and the higher academic education years of the students in the present study.

**Table 1. Demographic and background characteristics of the students.**

| Variable | | N | % |
|---|---|---|---|
| Gender | Female | 123 | 57.5 |
| | Male | 91 | 42.5 |
| Academic years | First Year | 79 | 36.9 |
| | Second Year | 56 | 26.2 |
| | Third Year | 41 | 19.2 |
| | Fourth Year and Above | 38 | 17.8 |
| Marital status | Single | 191 | 89.3 |
| | Married | 22 | 10.3 |
| | Consult a Physician and Phone Call | 47 | 21.9 |
| Ways to receive health content | Internet | 123 | 57.5 |
| | TV and Radio | 44 | 20.6 |
| Father's job | Employed | 139 | 65 |
| | Retired | 43 | 20.1 |
| | Other | 32 | 15 |
| Mother's job | Employed | 37 | 17.3 |
| | Housewife | 177 | 82.7 |
| Father's education | Non- University | 139 | 65 |
| | University | 75 | 35 |
| Mother's education | Non-University | 166 | 77.6 |
| | University | 48 | 22.4 |
| Smoking | Yes | 21 | 9.8 |
| | No | 193 | 90.2 |
| Hookah consumption | Yes | 37 | 17.3 |
| | No | 177 | 82.7 |
| Location | Dormitory | 169 | 79 |
| | Other | 45 | 21 |
| University Degree | Associate Degree | 27 | 12.6 |
| | Bachelor Degree | 187 | 87.4 |
| The amount of physical activity per week | Low | 118 | 55.1 |
| | Medium | 68 | 31.8 |
| | Much | 23 | 10.7 |
| Major | Clinical | 128 | 54.8 |
| | Not-Clinical | 86 | 40.2 |
| COVID-19 Infection Type | Omicron | 54 | 25.2 |
| | Other | 41 | 19.2 |
| | Don't Know | 65 | 30.4 |
| | I was not infected | 54 | 25.2 |

The results also showed that the adoption of preventive behaviors against COVID-19 was at a moderate level. Considering the average levels of health literacy in the present study, and the relationship between health literacy and the adoption of preventive behaviors [22–24,26,27], these results were justifiable too. However, in Nasirzadeh et al.s' study, the adoption of preventive behaviors against COVID-19 was at a high level [5] which could be due to the difference in the scale used for measuring preventive behaviors against COVID-19 and the different participants in the two studies. Also, the findings were consistent with the results of Panahi et al. [23], but it was not consistent with the results of Amjadian et al. [28] and the next study of

**Table 2. Frequency distribution of the health literacy level and behaviors to prevent COVID-19 among students.**

| Health Literacy Groups | N | % |
|---|---|---|
| Inadequate health literacy | 11 | 5.1 |
| Not enough health literacy | 53 | 24.8 |
| Adequate health literacy | 109 | 50.9 |
| Excellent health literacy | 40 | 18.7 |
| **Behaviors to prevent COVID-19** | **Frequency** | **Frequency Percentage** |
| Weak | 20 | 9.3 |
| Medium | 86 | 40.2 |
| Good | 108 | 50.5 |

Panahi et al. [22,24]. One of the reasons for this inconsistency was the difference in the topics and types of preventive behaviors between in these studies and the present one. Also, another possible reason could be that the participants were adolescents in the studies of Amjadian et al. [28] and Panahi et al. [22]. Also, in the current study, among the preventive behaviors against infection with COVID-19, washing hands with soap and water or disinfectants was the most frequent, and the behavior of using masks and gloves when leaving the house was the least frequent. These findings were similar to the results of the study by Khazai Pool et al. [4].

The results also showed that gender was one of the factors affecting the adoption of preventive behaviors. The adoption of preventive behaviors was higher in female students than in male ones because women follow health principles and medical recommendations, and they are more interested in learning health information than men do [17]. In Nakayama et al. [29], Raiserka et al. [30], Panahi et al. [31], and Panahi et al.s' studies [24] adoption of preventive behaviors against COVID-19 was reported to be more in women which were in line with our results.

Moreover, the results showed that physical activity was one of the factors influencing the adoption of preventive behaviors because doing physical activity in itself was a kind of preventive behavior. Therefore, the homogeneity of these two variables could probably justify the significant relationship between them. In Panahi et al. [23], and Amjadian et al. [28], there was a significant relationship between physical activity and the adoption of preventive behaviors which was consistent with the results here.

The results also showed that the mother's education level was one of the factors in adopting preventive behaviors. It can be said that possibly with the increase in the education level of the mothers, it is more probable for them to work outside. As a result, they go out more and their possible fear of being infected with COVID-19 is reduced. Therefore, they probably follow preventive behaviors less, and their children also probably follow their parents in adopting preventive behaviors, and as a result, they adopt such behaviors less too.

In the study of Raiserka et al. [30], there was also a significant relationship between the mother's education level and the adoption of preventive behaviors against COVID-19. The results of Etihad Nejad et al. [32], Manshadi et al. [33], Khani Jihouni et al. [34], and Panahi et al. [35] were also consistent with our results here.

The results also showed that the level of education was one of the factors affecting the adoption of preventive behaviors. There was a significant relationship between the level of education and adopting preventive behavior against COVID-19 in the study of Nguyen et al. [36]. However, the results of Panahi et al. [24,35] indicated that the number of academic years in students did not affect adopting preventive behaviors among them. This could be because of the difference in the preventive behaviors measured between the two studies.

**Table 3. Effective factors for adopting preventive behaviors in COVID-19 in the logistic regression model.**

| Variables | | Odds Ratio | CI | P-value |
|---|---|---|---|---|
| | Insufficient | 0.10 | (0.02–0.47) | **<0.0001** |
| Health literacy | Not quite enough | 0.34 | (0.13–0.87) | **0.02** |
| | Enough | 0.53 | (0.23–1.22) | 0.13 |
| | Excellent | 1 | | |
| Gender | Male | 0.25 | (0.12–0.53) | **<0.0001** |
| | Female | 1 | | |
| Mother's education | non-university | 2.26 | (1.04–4.92) | **0.03** |
| | University | 1 | | |
| Section | Associate degree | 3.02 | (1.10–8.29) | **0.03** |
| | Bachelor Degree | 1 | | |
| Smoking | Yes | 0.30 | (0.10–0.90) | **0.03** |
| | No | 1 | | |
| Amount of physical activity | High | 0.24 | (0.08–0.67) | **<0.0001** |
| | Moderate | 0.43 | (0.22–0.84) | **0.01** |
| | Low | 1 | | |
| Academic years | Fourth and above | 0.81 | (0.29–2.27) | 0.69 |
| | Third | 0.57 | (0.21–1.55) | 0.27 |
| | Second | 0.55 | (0.23–1.33) | 0.18 |
| | First | 1 | | |
| Marital status | Married | 0.81 | (0.28–2.26) | 0.70 |
| | Single | 1 | | |
| Father's job | Other | 1.06 | (0.42–2.68) | 0.90 |
| | Retired | 1.06 | (0.90–2.53) | 0.89 |
| | Employed | 1 | | |
| Mother's job | Employed | 0.88 | (0.32–2.44) | 0.82 |
| | Housewife | 1 | | |
| Father's education | non-university | 0.56 | (0.25–1.23) | 0.15 |
| | University | 1 | | |
| Location | non-dormitory | 0.90 | (0.39–2.10) | 0.81 |
| | a dormitory | 1 | | |
| Major | non-clinical | 1.48 | (0.70–3.10) | 0.30 |
| | Clinical | 1 | | |
| Hookah Consumption | Yes | 0.64 | (0.24–1.72) | 0.38 |
| | No | 1 | | |
| Ways to receive health content | Ask the doctor and phone | 0.95 | (0.415–2.16) | 0.89 |
| | Radio and television | 1.18 | (0.52–2.64) | 0.70 |
| | Internet | 1 | | |
| Type of COVID-19 Infection | Other | 1.97 | (0.74–5.23) | 0.17 |
| | Omicron | 1.08 | (0.44–2.65) | 0.86 |
| | I do not know | 0.81 | (0.35–1.89) | 0.63 |
| | I did not get infected | 1 | | |

The results also showed that smoking status was one of the factors affecting the adoption of preventive behaviors. The adoption of preventive behaviors was more common among non-smoking students than the smoking ones, because not smoking was itself a kind of preventive

behavior and could be a background for adopting other preventive behaviors. These results were also consistent with the results of Panahi et al. [24].

Also, the results showed that health literacy was one of the effective factors in adopting preventive behaviors. This could be because health literacy is a collection of skills, abilities, and capacities in various dimensions. These skills and capacities were sometimes updated in acquiring and obtaining medical and health information, sometimes in reading them, sometimes in understanding them, sometimes in processing and interpreting them, and sometimes in decision-making and applying this information [37]. which could influence adopting the preventive behaviors [38]. Health literacy plays an essential role in promoting peoples' responsibility to maintain their health; in other words, health literacy- by improving people's awareness- could be considered one of the basic factors in adopting preventive behaviors [23]. In Nakayama et al.'s study, people with higher health literacy took more precautions to adopt preventive behaviors against COVID-19 [29]. Also, this finding was consistent with the results of Panahi et al. [22–24,26,27], Izadi-Rad et al. [39], and Nguyen et al. [40].

To our knowledge, the present study was the first study that evaluated the relationship between different levels of health literacy and the adoption of preventive behaviors against COVID-19 in Iran. The limitations included; the present study was done in associate and bachelor degree students of medical sciences. Therefore, the results cannot be generalized to students from other parts of the country and other age and student groups. Therefore, it is recommended to conduct it in different populations and groups (with varying ages, education, residential areas, and post-graduate degrees). Also, ignoring other dimensions related to health literacy, such as self-efficacy, communication, and calculation was another limitation of the study. Ignoring cultural contexts and skills such as speaking, listening, and having contextual and cultural knowledge of people was another limitation of this study because these skills should be examined when we measure health literacy. The self-reported data and the small number of the sample were other limitations of the study too.

## Conclusion

Adopting preventive behaviors against COVID-19 was lower among students with lower levels of health literacy, male students, students with less physical activity, students with illiterate mothers, undergraduate students, and finally smokers. Therefore, it is necessary to pay more attention to these students in designing educational programs. It is suggested to carry out more extensive studies to clarify the effect of health literacy on the adoption of preventive behaviors against COVID-19. Also, it seems necessary to design a special scale to measure health literacy in COVID-19 in all age groups. In addition, it is recommended to investigate the relationship between different health literacy skills and the adoption of preventive behaviors against COVID-19 in future studies, so that more effective and shorter measures can be taken in this regard.

## Supporting information

**S1 Data.**
(RAR)

## Acknowledgments

This study was carried out with the support of the Research and Technology Vice-Chancellor of Esfarayen Medical Sciences University and the cooperation of the students of this university.

The authors would like to express their gratitude to all the students and officials who helped the researchers in this study.

## Author Contributions

**Conceptualization:** Rahman Panahi, Zahra Ghorbanpour.

**Data curation:** Zahra Ghorbanpour.

**Formal analysis:** Rahman Panahi, Mohiadin Amjadian.

**Investigation:** Rahman Panahi.

**Methodology:** Bagher Moradi, Mohiadin Amjadian.

**Project administration:** Bagher Moradi, Fereshteh Eidy.

**Resources:** Fereshteh Eidy.

**Software:** Fereshteh Eidy.

**Supervision:** Zahra Ghorbanpour.

**Writing – original draft:** Rahman Panahi, Zahra Ghorbanpour, Bagher Moradi, Mohiadin Amjadian.

**Writing – review & editing:** Rahman Panahi, Mohiadin Amjadian.

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
