## [Decision Letter · Decision Letter 0]

3 Apr 2023

PONE-D-22-35718The relationship between health literacy and the adoption of covid-19 preventive behaviors: a cross-sectional study in IranPLOS ONE

Dear Dr. Amjadian,

Thank you for submitting your manuscript to PLOS ONE. After careful consideration, we feel that it has merit but does not fully meet PLOS ONE’s publication criteria as it currently stands. Therefore, we invite you to submit a revised version of the manuscript that addresses the points raised during the review process.

Two reviewers have evaluated the manuscript and their comments are presented below. The reviewers suggest that the English language used in the manuscript could be improved to enhance the clarity of the scientific content. Additionally, they have requested clarification on whether the questionnaire used in the study was previously validated, and have recommended a thorough check of the accuracy and consistency of numerical values in the tables.

Could you please carefully revise the manuscript to address all comments raised?

We look forward to receiving your revised manuscript.

Kind regards,

Lucinda Shen, MSc

Staff Editor

PLOS ONE

Journal Requirements:

Reviewers' comments:

Reviewer's Responses to Questions

**Comments to the Author**

1. Is the manuscript technically sound, and do the data support the conclusions?

Reviewer #1: Yes

Reviewer #2: Partly

2. Has the statistical analysis been performed appropriately and rigorously? 

Reviewer #1: Yes

Reviewer #2: Yes

3. Have the authors made all data underlying the findings in their manuscript fully available?

Reviewer #1: Yes

Reviewer #2: No

4. Is the manuscript presented in an intelligible fashion and written in standard English?

Reviewer #1: Yes

Reviewer #2: No

5. Review Comments to the Author

Reviewer #1: Well-written paper. However, some sections need to be revised. The methods section should be more concise, and the results should reflect stated ways of reporting logistic regressions. The rationale behind health literacy for COVID-19 prevention behaviors should be well outlined in the discussion, especially if medical students have average health literacy. How does this impact health communication with patients or the general population? This manuscript has the potential to impact public health programming. Also, cross check grammar for errors. See comments below. Good job!

Reviewer #2: English language text should be checked and improved.

The introduction is inadequate, it just repeats the same thoughts without explaining the terms and without providing some background literature.

In the Method section they present the content of the ‘Demographic and background information’ of which the final element is “the type of Covid-19”. What does it mean here? Who’s type, what types exist at all, etc. This should be clarified!

In Table 1, the last column contains percentage of the participants answering to the particular question; however, the sum of the percentages regularly is 98 and not 100. Why?

The format applied in these tables are curious and unusual (e.g. 0/10, 0/34, 0/54). What this format means?

The performance percentages (%) at the Findings chapter (in most of the journals usually entitled as Results) are very confusing. These behaviors (what behaviors?) are mentioned as done less, in other sentences more, but it is not clear less of what?

In general, this paper should be significantly improved, regarding terms, data presentation, format and language.

6. PLOS authors have the option to publish the peer review history of their article (what does this mean?). If published, this will include your full peer review and any attached files.

Reviewer #1: No

Reviewer #2: No

---

## [Author Response · Author response to Decision Letter 0]

19 May 2023

Reviewer Response to the comments of the dear Reviewers

1. There are many types of coronavirus diseases, so it is important to state the specific disease. Done 

 ( Highlighted in Yellow in the manuscript)

2. Write COVID-19 in full first Done

( Highlighted in Yellow in the manuscript)

3. It may be helpful to indicate the country. (Consider using the third person). Done

( Highlighted in Yellow in the manuscript)

4. Was it a “sudden outbreak” worldwide? Consider revising this to more accurate information. Done

( Highlighted in Yellow in the manuscript)

5. Who were the participants in Mazandaran? Are these different from the 30 university students? Were these two different pilot studies? Were these studies separate from the current study?

 The participants in the previous study, conducted in Mazandaran, were people living in Mazandaran- a province in Iran. The designers of this instrument approved its reliability through a pilot study among 30 participants from Mazandaran.

Considering that students were studied instead of ordinary people in the current study, the current research team approved the reliability of this instrument for students by conducting a pilot study among 30 students. Therefore, these two pilot studies are different and separate from each other.

6. Revise this sentence. Who received the code of ethics?

 The code of ethics was obtained from the Research and Technology Vice-Chancellor of Esfrain Medical Sciences University by the main researcher ( The Corresponding Author).

7. What was the nature of informed consent? Written or verbal? Written informed consent was obtained from the students

8. Enter? Dear Reviewer; As we have mentioned in the manuscript, the variables were entered simultaneously. 

9. While some information in the methods section is helpful, the authors should consider being concise but comprehensive. As it is, it is difficult to follow the methodology logic/flow It was done and some information that seemed to be extra was deleted.

10. Consider writing the frequency as percentages. E.g. 57.5% rather than 57/5 Done

( Highlighted in Yellow in the manuscript)

11. It is unclear what chance ratio means. Done 

( Highlighted in Yellow in the manuscript)

12. Highlight in Table 3 any significant associations. Done

( Highlighted in Yellow in the manuscript)

13. Use standardized ways of reporting logistic regression results. E.g. Odds ratio, 95%CI, Done

( Highlighted in Yellow in the manuscript)

14. The means and standard deviations are not reflected in the tables. Ensure that what is presented in text and table match. Qualitative results were reported in the tables, while quantitative results were reported in the text.

15. n general, the tables are not synchronized. For example, Table 1 has University degree (Bachelor’s / Associate's) _ yet table 3 talks about section as (Associate and master's). Need to be consistent throughout. Done 

( Highlighted in Yellow in the manuscript)

16. Average literacy levels are based on what measure? What is considered high vs. low? 

As mentioned in the article, according to the designers of the questionnaire, scores from 0 to 50 are insufficient, 50.1 to 66 not very sufficient, 66.1 to 84 sufficient, and 84.1 to 100 are excellent health literacies. 

The average health literacy in the present study was 72, which, according to the research team, was at an average level. In other words, we believed that 72 was at a sufficient level and belonged to the average category. If the average health literacy was between 84.1 and 100, it would be in a good category. Also, if the average health literacy were between 0 and 66, then it would be placed in the weak category.

17. Why is it justifiable that medical students have average health literacy levels, yet they are expected to have higher levels generally?

 It was modified in the manuscript:

Considering the effect of the level of education on the level of health literacy of people, and the fact that almost two-thirds of the students in this study were in the first and second years of education, these results could be justified to some extent.

---

## [Decision Letter · Decision Letter 1]

28 Jun 2023

PONE-D-22-35718R1The relationship between health literacy and the adoption of covid-19 preventive behaviors: a cross-sectional study in IranPLOS ONE

Dear Dr. Mohiadin Amjadian

Thank you for submitting your manuscript to PLOS ONE. After careful consideration, we feel that it has merit but does not fully meet PLOS ONE’s publication criteria as it currently stands. Therefore, we invite you to submit a revised version of the manuscript that addresses the points raised during the review process.

There was some improvement from the previous submission. However, we invite you to make minor revisions , double check your work and ensure that all comments are addressed and that there are no errors. While some reviewers had differing comments, we believe that the title is appropriate and that your article is still valid for this journal. We apologize for taking long to respond. This was due to a lack of reviewers.  

Kind regards,

Sylvia Ayieko, 

Guest Editor

PLOS ONE

Journal Requirements:

Additional Editor Comments:

Dear Author,

Congratulations. The paper is quite good but needs a few revisions. See below.

Introduction

1. Recommend writing COVID-19 in all caps throughout the text.

2. Be consistent. Will you be using SARS-COV-2 or COVID-19, or both? If using COVID-19, write it in full in the first instance, then put COVID-19 in parenthesis.

3. Move line 87: The definition of Health literacy should be before explaining its effectiveness etc.

4. Lines 91-94: Does low health literacy cause diseases? Was there a causal effect? Consider changing the wording to an association or a relationship.

5. Line 90/95: Since you mention studies… provide multiple citations to support the claim.

6. Lines 105-107: need to have more substance. What does average mean?

7. Revise lines 112- 116. This paragraph can be comprehensive by not repeating health literacy in one sentence.

Methods

1. Line 116- Provide more information about Esfrain Faculty of Medical Sciences. What City and country?

2. Line 128: Revise this section. Briefly mention the three instruments at the beginning, then proceed to describe the instruments.

3. Line 133-158: The health literacy questionnaire's description is lengthy but can be made clearer. For example, rather than writing out each study that indicates the validity of the scale, simply just state that the questionnaire has been validated by (X, Y, and Z). Also, you could just note that the Likert scale ranges from 5 (completely easy )to 1 (completely difficult). You could also insert Cronbach's alpha coefficient at the beginning next to each item, e.g., Access (α=0.85), instead of repeating the items at the end of the paragraph.

4. 159-175- Condense this information as suggested above. Too long. You don’t have to mention that reliability was measured using a pilot study; simply cite them if describing reliability and validity.

Findings

1. Line 188: Suggest revision- Start with a word rather than a numeral. Also, consider stating that the participants completed the survey. “Studied” is a bit vague. What does it mean?

2. Table 2: Bold the “ Behaviors to prevent corona”. Be consistent in the description of COVID-19. Replace “corona”

3. Table 3: What do the figures in bold mean? Include more information as a description at the end of the table. Stick to two or three decimal points throughout.

4. Table 3: Under health literacy is “Enough“ significant? Is there a reason it is bolded?

5. The results section should be the core of the paper. You need to include additional information based on the analysis. What do the odds ratios mean based on the variables?

Reviewers' comments:

Reviewer's Responses to Questions

**Comments to the Author**

1. If the authors have adequately addressed your comments raised in a previous round of review and you feel that this manuscript is now acceptable for publication, you may indicate that here to bypass the “Comments to the Author” section, enter your conflict of interest statement in the “Confidential to Editor” section, and submit your "Accept" recommendation.

Reviewer #3: All comments have been addressed

Reviewer #4: All comments have been addressed

2. Is the manuscript technically sound, and do the data support the conclusions?

Reviewer #3: No

Reviewer #4: Yes

3. Has the statistical analysis been performed appropriately and rigorously? 

Reviewer #3: No

Reviewer #4: I Don't Know

4. Have the authors made all data underlying the findings in their manuscript fully available?

Reviewer #3: (No Response)

Reviewer #4: Yes

5. Is the manuscript presented in an intelligible fashion and written in standard English?

Reviewer #3: No

Reviewer #4: Yes

6. Review Comments to the Author

Reviewer #3: hi

Descriptive title is not suitable for this journal

The article is poorly written

A proper analysis has not been done

Reviewer #4: Congratulations on your valuable work. Comments are included to improve your article.

Abstract

Also report health literacy results in the results section.

Introduction

In the introduction section, line number 85, first include the definition of health literacy, then include previous studies related to health literacy and healthy behaviors.

Having or not having the disease of Covid-19 and having or not having the previous experience of suffering from the disease of Covid-19 are not among the entry criteria?

Method

In the method section, Cronbach's alpha include the different dimensions of the health literacy questionnaire.

In the method section of line number 180, specify which score is less than 50% and which score is more than 50%. The cut off point is not known.

Discussion

Information about study 12, 24, and 25 has not been included that these studies were conducted on whom and at what setting and with what tools.

Clarify the differences between 2 studies 5 and 28 with the current study.

7. PLOS authors have the option to publish the peer review history of their article (what does this mean?). If published, this will include your full peer review and any attached files.

Reviewer #3: No

Reviewer #4: No

---

## [Author Response · Author response to Decision Letter 1]

19 Jul 2023

Additional Editor Comments Point by point Responses to the comments

Recommend writing COVID-19 in all caps throughout the text. OK 

Be consistent. Will you be using SARS-COV-2 or COVID-19, or both? If using COVID-19, write it in full in the first instance, then put COVID-19 in parenthesis. OK 

Move line 87: The definition of Health literacy should be before explaining its effectiveness etc. Dear Reviewers: In response to your opinion regarding the weakness and inadequacy of the introduction, the research team believes that the introduction should be written according to the keywords in the title of the article. Therefore, in the first paragraphs, we talked about the corona virus and its complications and risk of infection and death. Then it was briefly talked about how to prevent it. Then, we went to health literacy, which is one of the effective factors in preventing Covid-19, and it is related to the adoption of preventive behaviors. Then, the definition of health literacy and the scope of its effects on people's health have been presented. In the following, we have addressed our other key word, students, and we have brought why students were chosen, and also their level of health literacy. Finally, the reason why the present study was done has been summarized.

Lines 91-94: Does low health literacy cause diseases? Was there a causal effect? Consider changing the wording to an association or a relationship. OK

Line 90/95: Since you mention studies… provide multiple citations to support the claim. OK

Lines 105-107: need to have more substance. What does average mean? OK

Revise lines 112- 116. This paragraph can be comprehensive by not repeating health literacy in one sentence. OK 

Line 116- Provide more information about Esfrain Faculty of Medical Sciences. What City and country? OK

Line 128: Revise this section. Briefly mention the three instruments at the beginning, then proceed to describe the instruments. OK

Line 133-158: The health literacy questionnaire's description is lengthy but can be made clearer. For example, rather than writing out each study that indicates the validity of the scale, simply just state that the questionnaire has been validated by (X, Y, and Z). Also, you could just note that the Likert scale ranges from 5 (completely easy) to 1 (completely difficult). You could also insert Cronbach's alpha coefficient at the beginning next to each item, e.g., Access (α=0.85), instead of repeating the items at the end of the paragraph. OK

159-175- Condense this information as suggested above. Too long. You don’t have to mention that reliability was measured using a pilot study; simply cite them if describing reliability and validity. OK 

Line 188: Suggest revision- Start with a word rather than a numeral. Also, consider stating that the participants completed the survey. “Studied” is a bit vague. What does it mean? OK 

Table 2: Bold the “ Behaviors to prevent corona”. Be consistent in the description of COVID-19. Replace “corona” OK 

Table 3: What do the figures in bold mean? Include more information as a description at the end of the table. Stick to two or three decimal points throughout. OK 

Table 3: Under health literacy is “Enough“ significant? Is there a reason it is bolded? OK

The results section should be the core of the paper. You need to include additional information based on the analysis. What do the odds ratios mean based on the variables? OK 

Reviewer #3: 

Descriptive title is not suitable for this journal

The article is poorly written

A proper analysis has not been done Dear Reviewer; We sought to determine factors influencing the adoption of preventive behaviors against Covid-19. Therefore, to achieve this goal, we used logistic regression, which is a cumulative statistical test that measures the effects of all variables simultaneously. In addition, we believe that if only health literacy was included in the regression model, then our study would be of the correlation type, while it is practically impossible to investigate this relationship without considering other variables. Therefore, in our opinion, a descriptive title is more appropriate. At the same time, we re-examined our analysis. We have strengthened the article as much as possible.

Reviewer #4: 

Congratulations on your valuable work. Comments are included to improve your article. Also report health literacy results in the results section. Dear Reviewer: Health literacy results have already been reported in the results section. 

In the introduction section, line number 85, first include the definition of health literacy, then include previous studies related to health literacy and healthy behaviors. Dear Reviewers: In response to your opinion regarding the weakness and inadequacy of the introduction, the research team believes that the introduction should be written according to the keywords in the title of the article. Therefore, in the first paragraphs, we talked about the corona virus and its complications and risk of infection and death. Then it was briefly talked about how to prevent it. Then, we went to health literacy, which is one of the effective factors in preventing Covid-19, and it is related to the adoption of preventive behaviors. Then, the definition of health literacy and the scope of its effects on people's health have been presented. In the following, we have addressed our other key word, students, and we have brought why students were chosen, and also their level of health literacy. Finally, the reason why the present study was done has been summarized.

Having or not having the disease of Covid-19 and having or not having the previous experience of suffering from the disease of Covid-19 are not among the entry criteria? NO

In the method section, Cronbach's alpha includes the different dimensions of the health literacy questionnaire. OK

In the method section of line number 180, specify which score is less than 50% and which score is more than 50%. The cut off point is not known. It was Corrected and strengthened. 

Information about study 12, 24, and 25 has not been included that these studies were conducted on whom and at what setting and with what tools.

Clarify the differences between 2 studies 5 and 28 with the current study. It was Corrected and strengthened.

---

## [Decision Letter · Decision Letter 2]

15 Sep 2023

PONE-D-22-35718R2The relationship between health literacy and the adoption of COVID-19 preventive behaviors: a cross-sectional study in Iran.PLOS ONE

Dear Dr. Mohiadin Amjadian,

Thank you for submitting your manuscript to PLOS ONE. After careful consideration, we feel that it has merit but does not fully meet PLOS ONE’s publication criteria as it currently stands. Therefore, we invite you to submit a revised version of the manuscript that addresses the points raised during the review process.

We look forward to receiving your revised manuscript.

Kind regards,

Hadi Ghasemi

Academic Editor

PLOS ONE

Journal Requirements:

Reviewers' comments:

Reviewer's Responses to Questions

**Comments to the Author**

1. If the authors have adequately addressed your comments raised in a previous round of review and you feel that this manuscript is now acceptable for publication, you may indicate that here to bypass the “Comments to the Author” section, enter your conflict of interest statement in the “Confidential to Editor” section, and submit your "Accept" recommendation.

Reviewer #4: (No Response)

Reviewer #5: All comments have been addressed

Reviewer #6: (No Response)

2. Is the manuscript technically sound, and do the data support the conclusions?

Reviewer #4: (No Response)

Reviewer #5: (No Response)

Reviewer #6: (No Response)

3. Has the statistical analysis been performed appropriately and rigorously? 

Reviewer #4: (No Response)

Reviewer #5: (No Response)

Reviewer #6: (No Response)

4. Have the authors made all data underlying the findings in their manuscript fully available?

Reviewer #4: (No Response)

Reviewer #5: Yes

Reviewer #6: (No Response)

5. Is the manuscript presented in an intelligible fashion and written in standard English?

Reviewer #4: (No Response)

Reviewer #5: Yes

Reviewer #6: (No Response)

6. Review Comments to the Author

Reviewer #4: Congratulations on your valuable research. Comments are included to improve your article.

Abstract

“Health literacy is one of the effective factors in controlling the 45 COVID-19 epidemic and is related to the adoption of preventive behaviors” When you bring this up in the background, it's like the issue is clear and no further research is needed. Revise the sentence.

Introduction

Line number 86 in the introduction section, define health literacy and describe its dimensions.

Of course, it is necessary to mention the dimensions of health literacy and its relationship with behaviors to prevent covid 19.

Include related and similar previous studies in this field in the background of the article and explain the necessity of your study.

Methods

Write the time frame of the research.

Explain what tests you used to analyze which variables

Discussion

Write the discussion more coherently. Write each finding of your study in a paragraph and after you compare it with previous studies, then write your conclusion.

Reviewer #5: Abstract

The mean and standard deviation (SD) of the scores of adoption of COVID-19 preventive behaviors and HL among students were 18.18±4.02 out of 25 and 72.14±12.75 out of 100, respectively.

What are the inclusion and exclusion criteria?

Please harmonize the statements. Make confused.

Adoption of COVID-19 preventive behaviors was less among students with lower levels of HL, male students, students with less physical activity, students with

educated mothers, undergraduate students, and smokers.

Please harmonize the statements.

Introduction

Sajjadi et al.'s

Azimi et al.'s

Need to explain what is Health literacy.

Correct this.( Sajjadi et al)

among whom 214 were selected by simple random so that the names of 115 all the students were listed and the participants were randomly selected.

Please harmonize this statement.

Health literacy for Iranian Adults (HELIA)

This questionnaire was adapted or adopted from where?

Results

Table 1& 2(Findings)

For the tables, write the symbol

% -percent

n-frequency

74.8 percent

Change to 74.8%

Discussion

The discussion is too descriptive.

How do the findings add to the body of scientific knowledge on the issue?

Recommendation:

Make the discussion based on subheadings, more clear to the reader.

E.g:

The relationship between the mother's education level and the adoption

of preventive behaviors.

Conclusion

Adoption of preventive behaviors against COVID-19 was lower among students with lower levels of health literacy, male students, students with less physical activity, students with

educated mothers, undergraduate students, and smokers.

Please harmonize this statement.

References

Most of the citations are above 5 years.

Very good.

Reviewer #6: Thank you for your effort and the fluent article. Here I provide some suggestion that may enhance the manuscript:

Line 55: you do not need to mention standard deviation. Just insert the mean score and insert SD in parentheses: 18.18 (±4.02)

Line 95: This seems more correct: University student population in Iran

Line 96: "Due to the important role, they play as managers and future planners in the country" omit the comma

Line 100: "Sajjadi et al.'s study showed that one-third of the students had insufficient or not very please for this and other studies mention the sample groups more specific.

Line 102: " The critical situation of the prevalence of COVID-19 disease can cause negative and positive psycho-social effects in society including the university students [14]" this sentence seems unnecessary. what is the relevance of this to your study? Did you consider these effects in your work?

Line 107-110: Was your sample group medical students or from other majors in health sciences? You may mention it here and it seems more reasonable when the participants are from health sciences.

The method section was described thoroughly and in details. Some comments:

How did you chose the background variables?

Line 113-116: "The statistical population was all the male and female students of Esfarayen Faculty of Medical Sciences, North Khorasan, Iran, among whom 214 were selected by simple random so that the names of all the students were listed and the participants were randomly selected." Please re-write this art in proper English writing.

Line 131: parents'

Line 133: what do you mean by "type of COVID-19"

LINE 179: please insert numbers alphabetically in the beginning of a sentence.

Table 1: what do you mean by "How to get content"- what content?

Table 1: it would be much better if you asked participants the weekly amount of their exercise and in some ranges ( e.g more than 2 hours, less than 2 hours …) this way it would be more objective.

Table 1: what is the importance of their Covid-19 type? What difference does it make?

Results: would you please a chart or graph ( as supplementary items) that shows the questionnaires items and the frequency of each answers in specific?

Discussion:

" These results were consistent with the results of Panahi et al. [24] and Panahi et al. [25]" please mention more details on these studies. Their sample. Their aim?

"In addition, the results of the present study were not consistent with the results of the study by Vozikis et al. [15] " what was this study results and what is the controversy?

In the discussion you could have compare the preventive behaviors in more detail with other studies. What type of behaviors were adapted the most? Which behaviors were neglected the most? It would be beneficial for future guidelines and educational purposes.

"The results also showed that gender was one of the factors affecting the adoption of preventive behaviors so the adoption of preventive behaviors was higher in female students than in male

ones because following health principles, and medical recommendations, and more interest in learning and obtaining health information are more in women than in men." Reference please.

" Mothers with higher education worked outside more and they go out more, thus their fear of COVID-19 was reduced. Therefore, they had probably fewer preventive behaviors and their children also followed their parents in adopting preventive behaviors." These statements are your opinion and your suggested reasons. But you have mentioned them like some facts. Please re-write this part or if they are proven in other studies please provide references.

Please provide a part on your suggestions or your studies implications.

7. PLOS authors have the option to publish the peer review history of their article (what does this mean?). If published, this will include your full peer review and any attached files.

Reviewer #4: No

Reviewer #5: **Yes: **DR RUSNANI AB LATIF

Reviewer #6: No

---

## [Author Response · Author response to Decision Letter 2]

8 Dec 2023

Comments Responses

Reviewer #4: The revised parts were highlighted in yellow inside the manuscript.

“Health literacy is one of the effective factors in controlling the 45 COVID-19 epidemic and is related to the adoption of preventive behaviors” When you bring this up in the background, it's like the issue is clear and no further research is needed. Revise the sentence. Tanks so much. It was revised. 

Line number 86 in the introduction section, define health literacy and describe its dimensions.

Of course, it is necessary to mention the dimensions of health literacy and its relationship with behaviors to prevent covid 19. 

Include related and similar previous studies in this field in the background of the article and explain the necessity of your study. 

 Dear Reviewers: the research team believes that the introduction should be written according to the keywords in the title of the article. Therefore, in the first paragraphs, we talked about the coronavirus and its complications and risk of infection and death. Then it was briefly talked about how to prevent it. Then, we went to health literacy, which is one of the effective factors in preventing COVID-19, and it is related to the adoption of preventive behaviors. Then, the definition of health literacy and the scope of its effects on people's health have been presented. In the following, we have addressed our other keyword, students, and we have explained why students were studied, and also their level of health literacy. Finally, the reason why the present study was done has been summarized.

Write the time frame of the research. Ok

Explain what tests you used to analyze which variables Ok

Write the discussion more coherently. Write each finding of your study in a paragraph and after you compare it with previous studies, then write your conclusion. 

 Dear reviewer; First, we started our discussion with the levels of health literacy and the adoption of preventive behavior, and then we analyzed each significant result completely. Please check again. Thanks.

Reviewer #5: 

The mean and standard deviation (SD) of the scores of adoption of COVID-19 preventive behaviors and HL among students were 18.18±4.02 out of 25 and 72.14±12.75 out of 100, respectively. Please harmonize the statements. Make confused. Thanks a lot. It was revised.

Adoption of COVID-19 preventive behaviors was less among students with lower levels of HL, male students, students with less physical activity, students with

educated mothers, undergraduate students, and smokers. Please harmonize the statements. 

Adopting preventive behaviors against COVID-19 was lower among students with lower levels of health literacy, male students, students with less physical activity, students with illiterate mothers, undergraduate students, and finally smokers.

Introduction

Sajjadi et al.'s

Azimi et al.'s

Need to explain what is Health literacy.

Correct this.( Sajjadi et al) It was revised.

In addition, health literacy was explained in the above lines inside the manuscript.

among whom 214 were selected by simple random so that the names of 115 all the students were listed and the participants were randomly selected.

Please harmonize this statement. The population included all male and female students at Esfarayen Medical Sciences University in Iran. The names of all students were listed and 214 participants were randomly selected by simple random sampling.

Health literacy for Iranian Adults (HELIA)

This questionnaire was adapted or adopted from where? 

This questionnaire was designed and psychometrically evaluated by Dr. Montazeri and colleagues in Iran and has the following specifications:

It consists of 33 items in 5 areas including access (6 items), reading skill (4 items), comprehension (7 items), evaluation (4 items), decision making, and use of health information (12 items). Was used. The scoring scale is Likert with 5 choices. In the items related to reading skills; 5 scores were assigned to the completely easy choice, 4 to the easy choice, 3 to neither the easy nor difficult choice, 2 to the difficult choices, and 1 to the completely difficult choice. Also, 5 scores were assigned to always, 4 to most of the time, 3 to sometimes, 2 to rarely, and 1 to never. The scoring method was such that the raw score of each participant in each area was obtained from the algebraic sum of the scores. To convert this score into a range of 0 to 100, the formula of the raw score difference is obtained from the minimum possible raw score divided by the difference between the maximum possible score and the minimum score. Then, to calculate the total score, the scores of all the dimensions (based on the range from zero to 100) are added and divided by the number of dimensions (5 dimensions). Scores from 0 to 50 are considered inadequate health literacy, 50.1 to 66 as insufficient health literacy, 66.1 to 84 as adequate health literacy, and 84.1 to 100 as excellent health literacy (18). This questionnaire has been designed and psychometrically evaluated by Montazeri and colleagues in Iran, and its validity and reliability have been confirmed. The construct validity in the 5 areas was 53.2%, and the reliability of the questionnaire items was determined to be 0.72 to 0.89 by Cronbach's Alpha Coefficient.

Table 1& 2(Findings)

For the tables, write the symbol

% -percent

n-frequency

74.8 percent

Change to 74.8% It was revised. 

The discussion is too descriptive. How do the findings add to the body of scientific knowledge on the issue? 

Dear reviewer, we believe that we have done a complete analysis, and all results have been analyzed, compared, and criticized with the existing studies in the discussion part.

To our knowledge, the present study is the first study that evaluated the relationship between different levels of health literacy and the adoption of preventive behaviors against COVID-19 in Iran. Therefore, it can add new findings to the existing scientific knowledge in this field.

Recommendation:

Make the discussion based on subheadings, more clear to the reader.

E.g:

The relationship between the mother's education level and the adoption

of preventive behaviors. All these things have been done and each significant relationship has been analyzed and compared.

Adoption of preventive behaviors against COVID-19 was lower among students with lower levels of health literacy, male students, students with less physical activity, students with educated mothers, undergraduate students, and smokers. Please harmonize this statement. 

Adopting preventive behaviors against COVID-19 was lower among students with lower levels of health literacy, male students, students with less physical activity, students with illiterate mothers, undergraduate students, and finally smokers.

References

Most of the citations are above 5 years.

Very good. Thanks so much. 

Reviewer #6: 

Line 55: you do not need to mention standard deviation. Just insert the mean score and insert SD in parentheses: 18.18 (4.02) Thanks a lot, it was revised. 

Line 95: This seems more correct: University student population in Iran Thanks a lot, it was revised. 

Line 96: "Due to the important role, they play as managers and future planners in the country" omit the comma Thanks a lot, it was revised.

Line 100: "Sajjadi et al.'s study showed that one-third of the students had insufficient or not very please for this and other studies mention the sample groups more specific. Thanks a lot, it was revised.

Line 102: " The critical situation of the prevalence of COVID-19 disease can cause negative and positive psycho-social effects in society including the university students [14]" this sentence seems unnecessary. what is the relevance of this to your study? Did you consider these effects in your work? Yes, we considered it.

We wanted to say that COVID-19 can even affect students.

Line 107-110: Was your sample group medical students or from other majors in health sciences? You may mention it here and it seems more reasonable when the participants are from health sciences. Esfrain Medical Sciences University has only 7 different fields of study including nursing, public health, intelligence, operating room technician, etc.

The method section was described thoroughly and in details. Some comments:

How did you chose the background variables? 

We chose the background variables by studying various studies in the field of health literacy and COVID-19.

Line 113-116: "The statistical population was all the male and female students of Esfarayen Faculty of Medical Sciences, North Khorasan, Iran, among whom 214 were selected by simple random so that the names of all the students were listed and the participants were randomly selected." Please re-write this art in proper English writing. 

The population included all male and female students at Esfarayen Medical Sciences University in Iran. The names of all students were listed and 214 participants were randomly selected by simple random sampling.

Line 131: parents' Thanks, it was revised. 

Line 133: what do you mean by "type of COVID-19" We meant the type of infection, such as O’micron, Gamma, Beta, etc.

LINE 179: please insert numbers alphabetically in the beginning of a sentence. Ok 

Table 1: what do you mean by "How to get content"- what content? It was revised. 

Table 1: it would be much better if you asked participants the weekly amount of their exercise and in some ranges (e.g more than 2 hours, less than 2 hours …) this way it would be more objective. 

Thanks a lot, your suggestion is very good, but the study has been completed and it can no longer be done.

Table 1: what is the importance of their Covid-19 type? What difference does it make? We wanted to see whether COVID-19 infection can be effective in preventing re-infection or initial infection.

Results: would you please a chart or graph (as supplementary items) that shows the questionnaires items and the frequency of each answers in specific? 

Thanks so much, but the results should be mentioned based on the most important goals of the study.

Discussion:

" These results were consistent with the results of Panahi et al. [24] and Panahi et al. [25]" please mention more details on these studies. Their sample. Their aim?

"In addition, the results of the present study were not consistent with the results of the study by Vozikis et al. [15] " what was this study results and what is the controversy? 

Both were done among students and we added to the relevant part.

Also, it is not appropriate to bring the goals in the discussion section.

In the discussion you could have compare the preventive behaviors in more detail with other studies. What type of behaviors were adapted the most? Which behaviors were neglected the most? It would be beneficial for future guidelines and educational purposes. Thanks, but this case is not a part of the objectives of this study.

"The results also showed that gender was one of the factors affecting the adoption of preventive behaviors so the adoption of preventive behaviors was higher in female students than in male

ones because following health principles, and medical recommendations, and more interest in learning and obtaining health information are more in women than in men." Reference please. 

The reference was given.

" Mothers with higher education worked outside more and they go out more, thus their fear of COVID-19 was reduced. Therefore, they had probably fewer preventive behaviors and their children also followed their parents in adopting preventive behaviors." These statements are your opinion and your suggested reasons. But you have mentioned them like some facts. Please re-write this part or if they are proven in other studies please provide references. It was revised.

Please provide a part on your suggestions or your studies implications. Dear reviewer, we have mentioned these things at the end of the results section:

It is suggested to carry out more extensive studies to clarify the effect of health literacy on the adoption of preventive behaviors against COVID-19. Also, it seems necessary to design a special tool to measure health literacy related to COVID-19 in all age groups.

.

---

## [Decision Letter · Decision Letter 3]

2 Jan 2024

PONE-D-22-35718R3The relationship between health literacy and the adoption of COVID-19 preventive behaviors: a cross-sectional study in Iran.PLOS ONE

Dear Dr. Amjadian,

Thank you for submitting your manuscript to PLOS ONE. After careful consideration, we feel that it has merit but does not fully meet PLOS ONE’s publication criteria as it currently stands. Therefore, we invite you to submit a revised version of the manuscript that addresses the points raised during the review process.

We look forward to receiving your revised manuscript.

Kind regards,

Hadi Ghasemi

Academic Editor

PLOS ONE

Journal Requirements:

Reviewers' comments:

Reviewer's Responses to Questions

**Comments to the Author**

1. If the authors have adequately addressed your comments raised in a previous round of review and you feel that this manuscript is now acceptable for publication, you may indicate that here to bypass the “Comments to the Author” section, enter your conflict of interest statement in the “Confidential to Editor” section, and submit your "Accept" recommendation.

Reviewer #4: All comments have been addressed

Reviewer #5: All comments have been addressed

Reviewer #6: (No Response)

2. Is the manuscript technically sound, and do the data support the conclusions?

Reviewer #4: Yes

Reviewer #5: Yes

Reviewer #6: Partly

3. Has the statistical analysis been performed appropriately and rigorously? 

Reviewer #4: I Don't Know

Reviewer #5: Yes

Reviewer #6: Yes

4. Have the authors made all data underlying the findings in their manuscript fully available?

Reviewer #4: Yes

Reviewer #5: Yes

Reviewer #6: No

5. Is the manuscript presented in an intelligible fashion and written in standard English?

Reviewer #4: Yes

Reviewer #5: Yes

Reviewer #6: Yes

6. Review Comments to the Author

Reviewer #4: (No Response)

Reviewer #5: Dear authors

Congratulation on the submitted manuscript. The topic is timely and will be of interest to the readers of the journal.

Thanks for the authors' effort on the paper. I have read and reviewed the manuscript, corrections from reviewers have been made.

Reviewer #6: Thank you for your efforts but some of my main comments were not answered properly. The suggestions were made to improve the final work but the authors neither made the changes nor provided reasonable justifications.

7. PLOS authors have the option to publish the peer review history of their article (what does this mean?). If published, this will include your full peer review and any attached files.

Reviewer #4: No

Reviewer #5: No

Reviewer #6: No

---

## [Author Response · Author response to Decision Letter 3]

19 Jan 2024

Comments Responses (The revised parts were highlighted in yellow inside the manuscript)

Line 102: "The critical situation of the prevalence of COVID-19 disease can cause negative and positive psycho-social effects in society including the university students [14]" this sentence seems unnecessary. what is the relevance of this to your study? Did you consider these effects in your work? Thanks a lot. It was revised and answered. (Fourth paragraph in the introduction part)

Results: would you please a chart or graph (as supplementary items) that shows the questionnaires items and the frequency of each answers in specific? Dear reviewer; Within the methodology, we have determined that the Helia questionnaire had 33 items, the behavior questionnaire included 5 items, and 15 items belonged to the background demographic variables. As we have already answered, the results should be mentioned based on the most important objectives of the study. The case considered by the respected reviewer is not within the scope of our goals, because we believe that we have mentioned the most important findings based on the study goals.

Discussion:" These results were consistent with the results of Panahi et al. [24] and Panahi et al. [25]" please mention more details on these studies. Their sample. Their aim?

…….

"In addition, the results of the present study were not consistent with the results of the study by Vozikis et al. [15]" what was this study results and what is the controversy? Both were done among students and we added to the relevant part. Also, it is not appropriate to bring the goals in the discussion section.

……………

In the discussion you could have compare the preventive behaviors in more detail with other studies. What type of behaviors were adapted the most? Which behaviors were neglected the most? It would be beneficial for future guidelines and educational purposes. Thanks, but this case is not a part of the objectives of this study. Thanks so much. It was revised and more details were brought into the manuscript.

Thanks so much. They were revised inside the manuscript.

"Mothers with higher education worked outside more and they go out more, thus their fear of COVID-19 was reduced. Therefore, they had probably fewer preventive behaviors and their children also followed their parents in adopting preventive behaviors."These statements are your opinion and your suggested reasons. But you have mentioned them like some facts. Please re-write this part or if they are proven in other studies please provide references. Thanks so much. They were revised in the manuscript.

Please provide a part on your suggestions or your studies implications. Dear reviewer:

 we have mentioned these things at the end of the conclusion:

It is suggested to conduct more extensive studies to clarify the effect of health literacy on the adoption of preventive behaviors against COVID-19 In addition, it is recommended to investigate the relationship between different health literacy skills and the adoption of preventive behaviors against COVID-19 in future studies, so that more effective and shorter measures can be taken in this regard. Also, it seems necessary to design a special tool to measure health literacy related to COVID-19 among all age groups.

---

## [Decision Letter · Decision Letter 4]

5 Feb 2024

The relationship between health literacy and the adoption of COVID-19 preventive behaviors: a cross-sectional study in Iran.

PONE-D-22-35718R4

Dear Dr. Mohiadin Amjadian,

We’re pleased to inform you that your manuscript has been judged scientifically suitable for publication and will be formally accepted for publication once it meets all outstanding technical requirements.

Kind regards,

Hadi Ghasemi

Academic Editor

PLOS ONE

Additional Editor Comments (optional):

Reviewers' comments:

Reviewer's Responses to Questions

**Comments to the Author**

1. If the authors have adequately addressed your comments raised in a previous round of review and you feel that this manuscript is now acceptable for publication, you may indicate that here to bypass the “Comments to the Author” section, enter your conflict of interest statement in the “Confidential to Editor” section, and submit your "Accept" recommendation.

Reviewer #6: All comments have been addressed

2. Is the manuscript technically sound, and do the data support the conclusions?

Reviewer #6: Yes

3. Has the statistical analysis been performed appropriately and rigorously? 

Reviewer #6: Yes

4. Have the authors made all data underlying the findings in their manuscript fully available?

Reviewer #6: Yes

5. Is the manuscript presented in an intelligible fashion and written in standard English?

Reviewer #6: Yes

6. Review Comments to the Author

Reviewer #6: Thank you, the revised version is sound and well-written and all the comments were properly answered.

7. PLOS authors have the option to publish the peer review history of their article (what does this mean?). If published, this will include your full peer review and any attached files.

Reviewer #6: No

---

## [Editor Report · Acceptance letter]

7 May 2024

PONE-D-22-35718R4 

PLOS ONE

Dear Dr. Amjadian, 

I'm pleased to inform you that your manuscript has been deemed suitable for publication in PLOS ONE. Congratulations! Your manuscript is now being handed over to our production team.

Kind regards, 

on behalf of

Dr. Hadi Ghasemi 

Academic Editor

PLOS ONE